# Collective dynamics of densely confined active polar disks with self- and mutual alignment

Weizhen Tang[1°], Yating Zheng[2,3°], Amir Shee[4], Guozheng Lin[5],
Zhangang Han[1], Pawel Romanczuk[2,3*] and Cristián Huepe[1,4,6†]

**1** School of Systems Science, Beijing Normal University,
Xinjiekouwaida Street 19, Beijing, 100875, China
**2** Department of Biology, Humboldt Universität zu Berlin,
Unter den Linden 6, Berlin, 10099, Germany
**3** Research Cluster of Excellence 'Science of Intelligence', Berlin, 10587, Germany
**4** Northwestern Institute on Complex Systems and ESAM,
Northwestern University, Evanston, IL 60208, USA
**5** School of Systems Science, Beijing Jiaotong University,
No.3 Shangyuancun, Beijing, 100044, China
**6** CHuepe Labs, 2713 West Augusta Blvd #1, Chicago, IL 60622, USA

⋆ pawel.romanczuk@hu-berlin.de , † cristian@northwestern.edu

## Abstract

We study the emerging collective states in a simple mechanical model of a dense group of self-propelled polar disks with off-centered rotation, confined within a circular arena. Each disk presents self-alignment towards the sum of contact forces acting on it, resulting from disk-substrate interactions, while also displaying mutual alignment with neighbors due to having its center of rotation located a distance $R$ behind its centroid, so that central contact forces can also introduce torques. The effect of both alignment mechanisms produces a variety of collective states that combine high-frequency localized circular oscillations with low-frequency milling around the center of the arena, in fluid or solid regimes. We consider cases with small/large $R$ values, isotropic/anisotropic disk-substrate damping, smooth/rough arena boundaries, different densities, and multiple systems sizes, showing that the emergent collective states that we identify are robust, generic, and potentially observable in real-world natural or artificial systems.

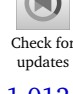

## Contents

---

° These authors contributed equally to the development of this work.

# 1 Introduction

Active systems are composed of individual components that consume energy to perform mechanical work [1–4]. They are inherently out-of-equilibrium and can self-organize into collective states that display, for example, flocking or phase-separation [5,6]. In most cases, activity is introduced in the form of self-propulsion and can lead to collective dynamics, as observed in various living and non-living systems [7–13]. Examples include collective motion in animal groups such as bird flocks [14,15], fish schools [16,17], herds of mammals [18,19], or insect swarms [20]; as well as in groups of bacteria [21–23] and of artificial agents such as vibrated polar disks [24, 25], colloidal particles [26, 27] or autonomous robots [28, 29]. The mechanisms that lead to the emergence of collective dynamics can be varied and complex. They can arise from different types of individual-level interactions, including: alignment [6,7], attraction [30], repulsion [31,32], directional consensus [33], speed synchronization [17], complex biological (neural, sensory, etc.) processes [34], social hierarchies [35], animal communication [36], escape-pursuit responses [37], or learning adaption [38].

Self-alignment is a minimal type of active dynamics that can lead to self-organization and is receiving increasing attention. It was the subject of a recent review paper [4] and can be found in a class of models where interactions only depend on relative positions (not on relative headings), often resulting from elastic forces [39–46]. Self-aligning dynamics can reach self-organized states, such as collective translation or rotation, through a mechanism that focuses the self-propulsion energy into low elastic modes. They can produce emerging collective motion states in elastically coupled tissue cells [39,40], collective actuation in active solids [46], collective dynamics in living active solids [45], and self-organized motion in a simple multi-cellular animal without a central nervous system [47]. Self-alignment has also been used in control algorithms for fixed formation drone swarms with only relative-position sensors [48].

In contrast to self-alignment, mutual alignment is defined as the tendency of interacting agents to directly match each other's headings, which requires the exchange of orientational information. This is the most commonly studied mechanism for flocking, at the basis of the Vicsek model and of other models with explicit polar angular alignment [6,12,49,50]. Agents with mutual alignment can self-organize into flocking states that result from a decentralized consensus process in the orientation. However, in solid or high-density systems where agents interact in a fixed lattice with the same neighbors over time, mutual alignment cannot achieve long-range order due to the Mermin-Wagner theorem [51], and can only result in flocking at finite scales, often in relatively small systems, as discussed in [10,12,52].

In realistic systems of densely packed dry active agents with elastic forces, we expect the interactions to combine self-alignment and mutual alignment effects. Indeed, any anisotropy in the agent-substrate interaction will introduce torques that can produce self-alignment, whereas any nonaxisymmetric feature in the agent shape with respect to its center of rotation will introduce torques between agents that can result in mutual alignment (even in the absence of inter-agent friction). This latter effect was explicitly considered in [31], for example, where self-propelled rods interacting through volume exclusion were shown to produce effective mutual alignment, because of their elongated shape. Recent studies on flocking in Janus colloids [27] and cooperative transport in anti-aligning robots [53] highlight the novel types of collective dynamics that can appear when considering anisotropic active agents.

A natural way to combine self-alignment and mutual alignment effects in a mechanical model is to consider off-centered attraction-repulsion forces that are not radial with respect to the center of rotation. The inter-agent forces then introduce torques that can result in mutual alignment, while simultaneously producing self-alignment. This approach was implemented in a minimal model of active solid introduced in [54], where a system of self-propelled agents connected by springs attached to lever arms was shown to display two types of order-disorder transition, one driven by the focusing of the self-propulsion energy into low elastic modes (produced by self-alignment) and the other by the decentralized heading consensus (resulting from mutual alignment). A similar approach was carried out to show that both types of self-organizing dynamics are also present in a minimal model describing densely packed self-propelled disks with linear repulsion only, each with an off-centered center of rotation, behind its geometrical center [55].

Motivated by these studies, in this work we will consider a densely packed system of self-propelled polar disks in circular confinement, each with its center of rotation at at distance $R$ behind its geometrical center. Here, $R$ becomes a control parameter that determines if the dynamics are dominated by self-alignment (for small $R$) or mutual alignment (for large $R$). For $R = 0$, the model displays only position-based, self-alignment interactions, as in [41,42]. Using this model, we will explore the boundary-mediated emerging collective states for cases with an either smooth or rough circular boundary, for isotropic or anisotropic agent-substrate damping, and for different system sizes. We note that most previous studies only considered periodic boundary conditions (for simplicity), despite the fact that introducing boundary effects is essential for connecting models to reality [56,57].

Our results show that a combination of two types of self-organized states can emerge, a state of localized rotation where the system displays small solid body circular oscillations (with all agents moving in small synchronized circles), and a milling vortex state where all agents orbit around the center of the arena. We find that the milling state consistently emerges for large $R$, irrespective of the type of boundary or agent-substrate damping. However, for small $R$, the boundary becomes key in shaping the emergent states; a milling vortex forms for smooth boundaries while localized rotation appears for rough boundaries. These collective states can coexist, with the first one becoming more prominent for small $R$, in cases dominated by self-alignment, and the second one for large $R$, when mutual alignment dominates. In

small systems the coexisting dynamics segregates into two distinct domains separated by a single orbiting vortex; a milling outer ring (driven by mutual alignment) and a central region displaying circular oscillations (driven by self-alignment). In larger systems the coexisting states becomes more disorganized, with multiple vortices, defects, and domains spontaneously appearing and disappearing. For lower densities, the center of the arena is always voided and a fluid milling state is favored. For larger densities, we find that solid body rotation and collective oscillations become more prominent.

Our work shows that dense systems of self-propelled agents with off-centered, nonaxisymmetric rotational dynamics can produce, both, self-alignment and mutual alignment processes, demonstrating the coexistence of two forms of self-organization in active systems. We expect experimental systems to be capable of displaying both types of self-organizing dynamics in real-world setups (since off-centered and nonaxisymmetric rotation will naturally occur if the interactions between neighbors and with the substrate are not perfectly axisymmetric), potentially resulting in a rich combination of emergent states.

## 2 Model setup

### 2.1 Dynamics of active polar disks with off-centered rotation

We describe a dense system of active elastic agents confined in a circular arena, to carry out simulations as detailed in Appendix A. Each agent is a soft disk $i$ with position $\vec{r}_i$, defined by its center of rotation that self-propels with preferred speed $v_0$ along the heading direction indicated by its orientation unit vector $\hat{n}_i = (\cos(\theta_i), \sin(\theta_i))$, as shown in Fig. 1a. All disks have the same diameter $l_0$ and a repulsive force appears between disks $i$ and $j$ when they overlap, given by $\vec{f}_{ij} = (k/l_0)(|\vec{l}_{ij}| - l_0)\vec{l}_{ij}/|\vec{l}_{ij}|$ if $|\vec{l}_{ij}| < l_0$, and by $\vec{f}_{ij} = 0$ otherwise, where $\vec{l}_{ij} = \vec{l}_j - \vec{l}_i$ with $\vec{l}_i = \vec{r}_i + R\hat{n}_i$. The total repulsive forces exerted by other agents on disk $i$ will thus be $\vec{F}_i = \sum_{j \in S_i} \vec{f}_{ij}$, where $S_i$ includes the indexes of all the neighbors of disk $i$. The combined effect of the interaction forces and self-propulsion leads to the following overdamped equation of motion for the position of each agent [4, 55]

$$\dot{\vec{r}}_i = v_0 \hat{n}_i + \mathbb{M}\vec{F}_i. \tag{1}$$

Here, we introduced the mobility matrix $\mathbb{M} = \mu_{\parallel}\hat{n}_i\hat{n}_i^T + \mu_{\perp}(\mathbb{I} - \hat{n}_i\hat{n}_i^T)$ to allow for anisotropic disk-substrate interactions, with different damping coefficients along ($\mu_{\parallel}$) and perpendicular ($\mu_{\perp}$) to the heading direction $\hat{n}_i$. We will focus on two limiting cases: (i) on the isotropic mobility case, where $\mu_{\parallel} = \mu_{\perp}(= \mu)$ and $\mathbb{M} = \mu\mathbb{I}$, with $\mathbb{I}$ representing $2 \times 2$ identity matrix, and (ii) on the fully anisotropic case, where $\mu_{\perp} = 0$ and $\mathbb{M} = \mu_{\parallel}\hat{n}_i\hat{n}_i^T$, which implies that agents can only move along the direction of $\hat{n}_i$, as in the case of wheeled robots. The isotropic limit describes disk-substrate interactions where external forces lead to the same displacement regardless of agent orientation, as in the case of drones [48] or legged robots [29], and the anisotropic limit corresponds to cases where agents cannot move sideways, as in the case of wheeled robots [43].

The disk-substrate interaction can lead to self-alignment, where the heading tends to align towards the sum of total forces on the disk. While self-aligning torques may originate from various mechanisms, as detailed in [4], we consider here that they have the form $(\hat{n}_i \times \vec{F}_i) \times \hat{n}_i = (\hat{n}_i \times \vec{F}_i) \cdot \hat{z} = (\vec{F}_i \cdot \hat{n}_i^{\perp})\hat{n}_i^{\perp} = (\mathbb{I} - \hat{n}_i\hat{n}_i^T)\vec{F}_i$, where $\hat{z}$ is the unit vector normal to the plane and $\beta$ controls the self-aligning strength. Note that these torques will depend on the details of the disk-substrate interaction. On the other hand, the position and center of rotation $\vec{r}_i$ of each disk $i$ is located a distance $R$ behind its geometrical center (with respect to its heading direction), at $\vec{l}_i = \vec{r}_i + R\hat{n}_i$, which introduces mutual alignment due to the torques

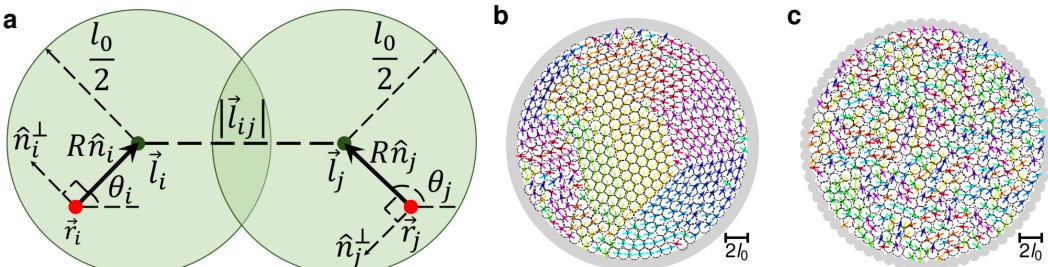

Figure 1: **Active polar disks with linear elastic repulsion model and arena. a**
Schematic diagram of two interacting self-propelled disks, $i$ and $j$, oriented along $\hat{n}_i$
and $\hat{n}_j$, respectively. Each disk has diameter $l_0$ and its center of rotation is labeled
by a red point, located a distance $R$ behind its geometrical center. Linear repulsive
forces, proportional to $|\vec{l}_{ij}| - l_0$, act on each centroid, resulting in forces and torques
on $\vec{r}_i$ and $\vec{r}_j$. **b** Snapshot of a relatively ordered densely packed group of active polar
disks in circular confinement, inside a smooth circular barrier introduced as a radial
linear outer potential. **c** Snapshot of a relatively disordered densely packed group of
active polar disks confined by a ring of fixed disks with the same repulsion potential
as the active disks, representing a rough boundary. Each arrow corresponds to an
active disk and its self-propulsion direction, colored by angle.

on the arm of length $R$ between the geometrical center (where the central forces are acting)
and the center of rotation. By combining these torques with the effects of noise, we find a
dynamical equation for the heading direction $\hat{n}_i$, given by

$$\dot{\hat{n}}_i = \beta(\mathbb{I} - \hat{n}_i \hat{n}_i^T)\vec{F}_i + \sqrt{2D_\theta}\eta_i(t)\hat{n}_i^\perp, \qquad (2)$$

where $\eta_i$ is a random variable that introduces Gaussian white noise with zero mean and vari-
ance $\langle \eta_i(t)\eta_j(t') \rangle = \delta_{ij}\delta(t - t')$. Here, $D_\theta$ sets the rotational noise strength and corresponds
to the rotational diffusion coefficient.

## 2.2 Simulation arena and parameters

We study the collective regimes that result from integrating Eqs. (1) and (2) for a dense group
of disks confined in a circular arena under different conditions. The simulation arena is defined
by a smooth or rough circular confinement barrier. In the smooth circular confinement case,
the outer ring boundary exerts a repulsive radial linear force inwards, perpendicular to its
surface, as shown in Fig. 1b. In the rough circular confinement case, the boundary is composed
of fixed disks with a repulsive potential equivalent to that of the active disks in the simulation,
as shown in Fig. 1c.

We consider homogeneous soft disks, all having the same diameter $l_0 = 1$ and self-
propulsion speed of $v_0 = 0.002$, unless otherwise noted. The force alignment strength co-
efficient is set to $\beta = 1.2$ and the simulation time step is $dt = 0.01$ in all simulations. All
simulations are carried out starting from random initial positions and orientations.

To characterize different scenarios, two values of the off-centered rotation distance $R$ are
chosen: $R = 0.03$ for weak mutual alignment $R = 0.3$ and for strong mutual alignment.
The $R$-dependent transition in vortex dynamics is further explored by varying $R$ from 0 to
0.3. We consider two limiting cases of mobility: a fully isotropic case with $\mu \equiv \mu_\parallel = \mu_\perp$
and $\mu = 0.02$, where the disk-substrate damping is uniform across all directions, and a fully
anisotropic damping case with $\mu_\perp = 0$ and $\mu_\parallel = 0.02$, where the agents can only move forward
or backward (as in the case of wheeled vehicles).

Most analyses focus on systems with $N = 400$ agents, while additional simulations with $N = 1600$ and $N = 6400$ agents are conducted to examine how the emergent states evolve in larger systems. The radius of the circular arena is set to $R_c = 10.5$ for $N = 400$ soft polar disks, leading to a packing fraction $\phi = N l_0^2 / 4 R_c^2 = 0.907$, where $l_0 = 1$ is the disk diameter. We vary the radius of the circular confinement, $R_c$, for different total numbers of active polar disks, $N$, to maintain constant packing fraction, $\phi = N l_0^2 / 4 R_c^2$, as shown in Fig. 6. For $N = 400, 1600, 6400$, we set $R_c = 10.5, 21, 42$ with $l_0 = 1$, all corresponding to $\phi = 0.907$.

A detailed definition of all the simulation variables, along with their specific values in our simulations, are provided in Table 1. The parameter lists used to generate the majority of the simulation results presented in this article. Any difference with respect to these values is explicitly noted in the figure captions.

## 3 Results

### 3.1 Overview of emergent states

We begin by providing an overview of the different regimes observed for agents with isotropic or anisotropic damping and small or large $R$, confined by smooth or rough boundaries. All simulations were carried out as detailed in Appendix A.

We characterize the emerging collective states using two order parameters. The first is the commonly used Polarization ($P$). It shows the degree of alignment and is defined by

$$P = \frac{1}{N} \langle \| \sum_{i=1}^{N} \hat{n}_i \| \rangle_t, \tag{3}$$

where $\langle \cdot \rangle_t$ represents the time average after reaching steady state. The second one measures the degree of milling in the system, and is defined as the milling order parameter

$$M = \frac{1}{N} \langle \| \sum_{i=1}^{N} \hat{r}_{ig} \times \hat{n}_i \| \rangle_t. \tag{4}$$

Here, $\hat{r}_{ig} = (\vec{r}_i - \vec{r}_g) / \| \vec{r}_i - \vec{r}_g \|$ points to the position of agent $i$ with respect to the center of the group $\vec{r}_g$. The order parameters $P$ and $M$ range from 0 to 1, where a disordered state will display low values for both, a flocking state will have high $P$ and low $M$, and a milling state will have low $P$ and high $M$.

Fig. 2 illustrates the collective states observed in simulations of 400 disks with diameter $l_0 = 1$, self-propulsion speed $v_0 = 0.002$, alignment strength $\beta = 1.2$, low noise $D_\theta = 0.001$, and time step $dt = 0.01$. Two types of agent-substrate damping are considered: isotropic damping with mobility coefficients $\mu \equiv \mu_\parallel = \mu_\perp$ and $\mu = 0.02$ (in panels a-d) and fully anisotropic damping with $\mu_\perp = 0$ and $\mu_\parallel = 0.02$ (in panels e-h). The four panels on the left side (a,b,e,f) correspond to smooth confinement, while the right panels (c,d,g,h) represent rough boundaries composed of inert, immovable disks. Two values for the off-centered rotational distance $R$ are explored: large ($R = 0.3$) in (a,c,e,g) and small ($R = 0.03$) in (b,d,f,h). The corresponding temporal dynamics are available as Supplementary Movies 1-8.

The different panels in Fig. 2 show that a *milling* state with high $M > 0.9$ and low $P \ll 1$ (where all agents orbit around a common center of rotation), will appear at large $R = 0.3$, for isotropic or anisotropic mobility and a smooth or rough boundary, as shown in panels (a,c,e,g). For small $R = 0.03$, in the smooth boundary case the vortex leaves the center of the arena and orbits in a circle at an approximately fixed distance from the border. Inside this orbit the agents are somewhat aligned and rotate mainly in place, while outside they continue

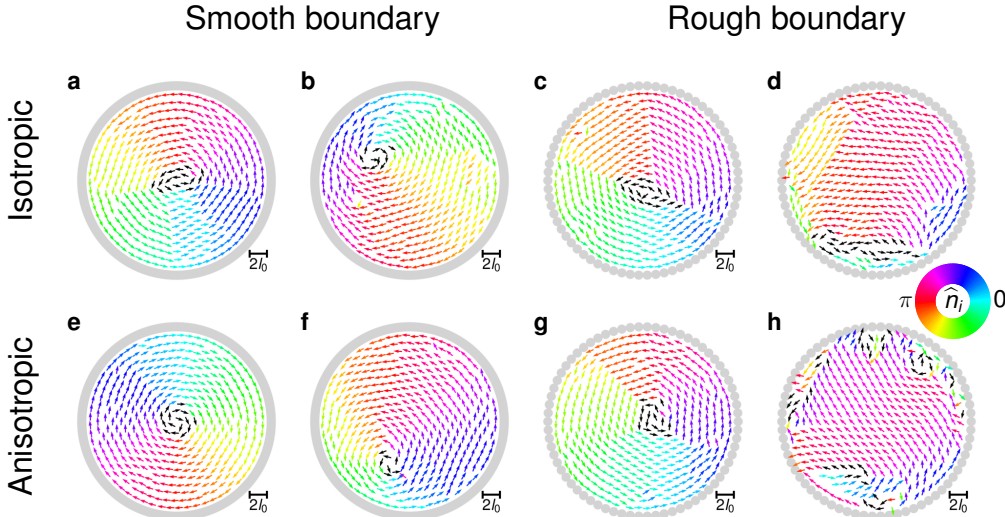

Figure 2: **Emerging collective states for different disk dynamics and boundary conditions.** Snapshots of collective states that emerge under different conditions, for the same $D_\theta = 0.001$ noise level. Each arrow, colored by angle, represents the location and orientation of a self-propelled disk. Black arrows signal high local vorticity. The top row panels (**a-d**) present cases with isotropic mobility, where agents can be displaced in any direction; the bottom row panels (**e-h**) correspond to fully anisotropic mobility, where they only move forward or backwards. The two left columns display simulations with a smooth boundary; the two right ones show cases with rough confinement. Panels **a**, **c**, **e**, **g** display the milling states observed for disks with a large off-centered rotation distance $R = 0.3$ (with respective polarization $P = 0.041, 0.135, 0.043, 0.086$ and milling order parameter $M = 0.974, 0.918, 0.971, 0.949$) Panels **b**, **f** show the combined outer milling and inner localized circular oscillation regions, separated by an orbiting vortex, obtained for disks with small $R = 0.03$ (with $P = 0.388, 0.414$ and $M = 0.753, 0.734$) in a smooth boundary confinement. Panels **d**, **h** show the localized circular oscillation state observed for disks with small $R = 0.03$ (with $P = 0.564, 0.653$ and $M = 0.459, 0.307$) in the rough boundary case.

milling with $M \sim 0.7$ and $P \sim 0.4$, as shown in panels (b,f). In the rough boundary case, the inner region appears well aligned, with each agent rotating round its own center of rotation, without significant displacements. Near the boundary, we find a layer of agents that are not aligned and display no milling, since they are trapped by the boundary roughness as shown in panels (d,h). This results in a larger polarization $P > 0.5$ and lower milling $M < 0.5$.

In addition to the three collective states described above, for higher values of noise $D_\theta$ and off-centered rotation distance $R$ we observed a standard state of dynamic disorder (where both $P$ and $M$ are very small and agents rotate locally in a disordered way) and a novel state of *quenched disorder* previously analyzed in [54, 55], both of which are shown in the phase diagrams displayed in Appendix B.

## 3.2 Local and global rotation modes

In order to explore the different rotational modes more in detail, we compute the orientation temporal autocorrelation function $C(\tau)$ to evaluate how the particles' $\hat{n}_i(t)$ orientations change over time and identify their rotational frequencies. We define this function for $N$ par-

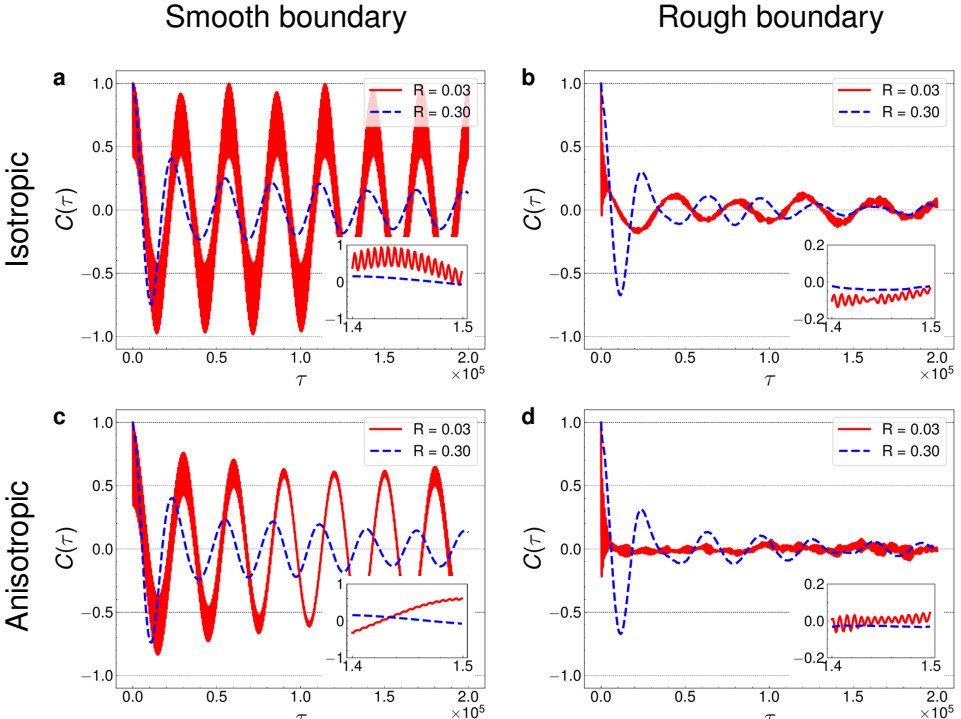

Figure 3: **Orientation autocorrelation functions for different collective states.**
The same cases presented in Fig. 2 are displayed, for smooth or rough arena boundaries and isotropic or anisotropic active disk mobility. The blue dashed curves correspond to disks with large off-centered rotation distance $R = 0.3$, whereas the red solid curves correspond to the small $R = 0.03$ case. The insets zoom into a short temporal interval to visualize high-frequency oscillations in the correlation functions, which only appear in the small $R$ case. Other simulation parameters are set to: $l_0 = 1.0$, $v_0 = 0.002$, $D_\theta = 0$, and $N = 400$.

ticles using the expression

$$C(\tau) = \langle \hat{n}_i(t) \cdot \hat{n}_i(t+\tau) \rangle_{N, N_S} . \tag{5}$$

Here, $\hat{n}_i(t)$ and $\hat{n}_i(t+\tau)$ are the orientation unit vectors at the time $t$ and $t+\tau$, respectively. The $\langle \cdot \rangle_{N, N_S}$ brackets represent the mean over all $N$ agents and all $N_S$ sampling points over time. Our orientation autocorrelation results are computed for runs lasting 20000 computational time units after reaching their stationary state, which we then sample every $\Delta t = 100$ to obtain $N_S = 2000$ sampling points.

We will use this tool to examine in detail the rotational dynamics for the same parameters used in the previous section, but with zero noise ($D_\theta = 0$), in order to visualize more clearly all persistent oscillations.

Figure 3 plots the orientation autocorrelation functions $C(\tau)$ for the same parameter combinations displayed in Fig. 2. Here, the large $R = 0.3$ case is displayed as blue dashed lines, and the small $R = 0.03$ case as red solid lines. It is notable that the amplitude of correlations decays rapidly in rough boundary setups (panels (b) and (d)), as disk-boundary interactions disrupt the temporal correlations of agents near the edge. We note, in addition, that the amplitude of the $C(\tau)$ curves in the small $R = 0.03$ case also decay more rapidly for systems with anisotropic damping, when compared to their isotropic counterparts (i.e., the amplitude of the solid red curves becomes smaller more rapidly in the anisotropic case). In the smooth boundary setups (panels (a) and (c)), angular dynamics are highly organized, displaying persistent

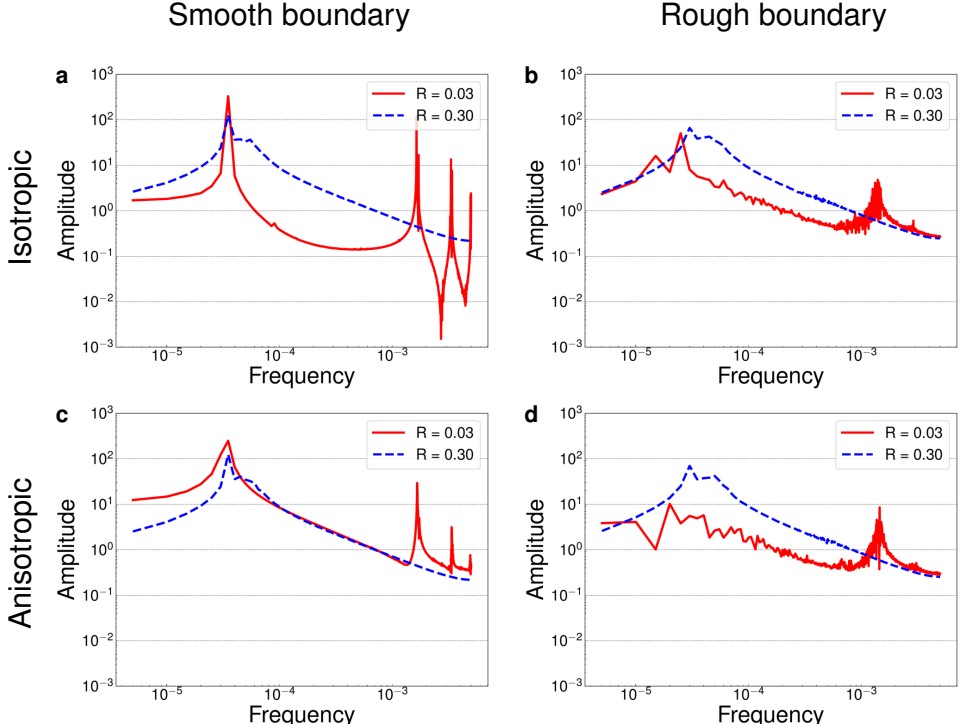

Figure 4: **Orientation autocorrelation in Fourier space for different collective states.** The same cases presented in Figs. 2 and 3 are displayed, for smooth or rough arena boundaries and isotropic or anisotropic active disk mobility. The blue dashed curves represent large off-centered rotation distance $R = 0.3$; the red solid curves correspond to small $R = 0.03$. Low frequency milling can be found to different degrees in all cases, but high-frequency circular oscillations of different intensities, with or without resonances, only appear in the small $R = 0.03$ case. Other simulation parameters are set to: $l_0 = 1.0$, $v_0 = 0.002$, $D_\theta = 0$, and $N = 400$.

rotations. Conversely, in rough boundary setups (panels (b) and (d)), temporal orientational correlations rapidly decay to low amplitude. Across all cases, anisotropic damping produces lower-amplitude correlation curves than isotropic damping. Notably, for small $R = 0.03$, red curves exhibit a superposition of high- and low-frequency oscillations, whereas large $R = 0.3$ cases (blue dashed lines) display only low-frequency oscillations. The orientation autocorrelation functions for a larger system size of $N = 1600$ are shown in Fig. 12.

Figure 4 presents the Fourier transforms of the correlation functions in Fig. 3 for the same parameter combinations, with the red lines representing again $R = 0.03$ and the dashed blue lines, $R = 0.3$. We observe that the low-frequency peaks (around $3 \times 10^{-5}$) correspond to the milling state (where the agents are orbiting around a common center of rotation). On the other hand, the sharper peaks above frequency $10^{-3}$ correspond to localized rotation states (where each agent is rotating around its own nearby center of rotation).

For small off-centered distance $R = 0.03$, Fig. 4 shows a low-frequency maximum for all combinations of smooth or rough boundary and isotropic or anisotropic disk-substrate damping. This maximum is most pronounced in the smooth isotropic case, where the lack of boundary perturbations or motion restrictions facilitates the formation of a solid rotating structure that forces the agents to turn with the same angular speed. It is the least pronounced in the rough anisotropic case, where these perturbations and restrictions favor the formation of a fluid rotating state, in which all the milling agents advance at near $v_0$ speed and thus the outer orbits have smaller angular speed than the inner ones. All the panels also display the high fre-

quency peaks that characterize localized rotation, suggesting that this state could be observed under a broad ange of conditions. In the smooth boundary case, the high-frequency peaks appear more sharply defined and reach higher values, showing that the lack of perturbations (introduced by boundary irregularities) results in the formation of groups of agents rotating with almost exactly the same frequency, and corresponding harmonics. Instead, in the rough boundary case, a single high-frequency peak appears that is less pronounced, more noisy.

For a large off-centered distance $R = 0.3$, Fig. 4 shows no high frequency peak, so the localized rotation state is fully suppressed, while still displaying the low-frequency maximum that characterizes the milling state. As in the small $R$ case, this maximum is more pronounced in simulations with a smooth boundary, which favor solid body rotation. In contrast to the small $R$ case, the maximum seems to be equally pronounced for isotropic or anisotropic damping, which is likely due to the fact that the motion of an agent with large $R$ is already constrained by external forces to be along its $\hat{n}_i$. The Fourier transform of orientation autocorrelation functions for a larger system size of $N = 1600$ are shown in Fig. 13. Finally, we found that the orientation autocorrelation function has the same frequency components in different regions in the bulk, so the rotational dynamics of the disks are essentially homogeneous throughout the arena until the boundary region is reached.

## 3.3  Vortex dynamics

The results presented above show that the localized rotation state and the milling state can coexist in some regimes. Moreover, for the parameters used in panels (b) and (d) of Fig. 2, the circular arena divides into two distinct regions: an outer ring exhibiting milling dynamics and a central circular area characterized by localized rotation, separated by a rotating vortex. We begin our analysis of this regime by examining the dynamics of this vortex in the anisotropic case, which is equivalent to its behavior in the isotropic case.

Figure 5 presents the vortex dynamics for the smooth boundary case displayed in Fig. 2f, with $R = 0.03$, $D_\theta = 0.001$, $N = 400$, $l_0 = 1.0$, and $v_0 = 0.002$. After identifying the vortex location, as detailed in Appendix C, its successive positions are traced every $\delta t = 100$ simulation time units ($10^4$ simulation time steps). Panel (a) depicts the vortex trajectory, showing a single vortex orbiting at an approximately constant rate and fixed distance from the center of the arena. Panel (b) shows that the corresponding angular positions match an orbital speed given by 0.0108 radians per simulation time unit.

When we extend our study to the rough boundary case and larger systems, we find that this simple, well structured regime with coexisting regimes separated by a single orbiting vortex becomes unstable. Instead, multiple vortices spontaneously appear and disappear in different parts of the arena. Fig. 6 displays the probability density function (PDF) of the radial position all vortices with respect to the center of the arena for $N = 400, 1600, 6400$ agents, keeping the density constant by increasing proportionally the size of the arena. The corresponding scatter plots of all identified vortex positions are displayed in Figs. 16 - 19.

We find that only the smooth boundary case with $N = 400$ agents present a single vortex in the arena (see Figs. 16 and 18). For $N = 1600$ we find between one and two vortices. For $N = 6400$ we always find more than two vortices in the arena, and the number depends on $R$. For $R = 0.3$, we find about two vortices in the arena in all cases, whereas for $R = 0$, we find close to 5 vortices on average for isotropic damping and 7 for anisotropic damping. In general, larger systems, low $R$ values, and anisotropic damping tend to produce more vortices. For larger $R$, the vortices tend to move towards the center of the arena, and for smaller $R$ they form at an approximately fixed distance from the edge.

In the rough boundary case, more vortices tend to nucleate near the boundary. As shown in Fig. 17 and  19, only the small $N = 400$ and large $R = 0.3$ cases produce one to two vortices per frame, since here vorticity is mainly driven by the global milling rotation. More vortices

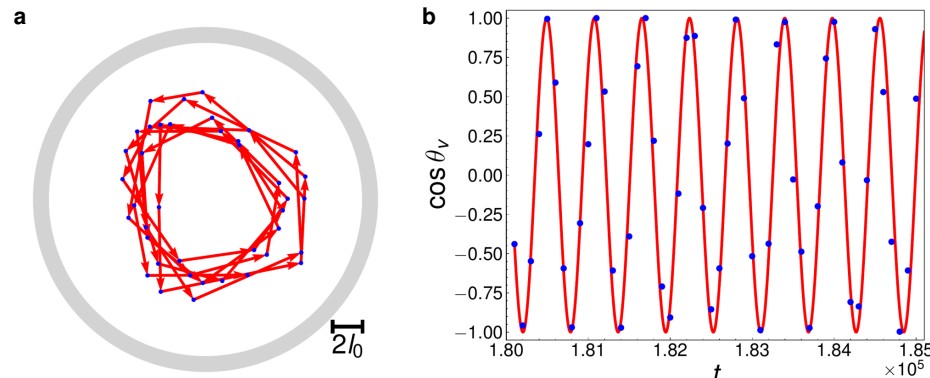

Figure 5: **Trajectory and dynamics of orbiting vortex.** The position of the orbiting vortex that appears in small $N = 400$ systems with a smooth boundary and small off-centered rotation distance $R = 0.03$ is displayed over time. **a** The trajectory of the vortex is traced in the arena by labeling its position every $\Delta t = 100$ computational time units as successive blue dots connected by red arrows. **b** Fitting of the cosine function of the angular position of the vortex as a function of time. The blue dots label again the angular position every $\Delta t = 100$, while the red curve is a sinusoidal fit given by $\cos \theta_v = \cos(0.0108 * t + 2.03)$. Other simulation parameters are set to: $\mu_\parallel = 0.02$ and $\mu_\perp = 0$ (imposing anisotropic damping), $l_0 = 1.0$, $v_0 = 0.002$, and $D_\theta = 0.001$.

appear in the arena for larger $N$ and smaller $R$, reaching an average of close to 13 and 26 vortices per frame for $N = 6400$ and $R = 0$, in cases with isotropic and anisotropic damping, respectively. Here again, more vortices are produced for large $N$ and small $R$. For $R = 0.3$ the vortices appear in the bulk of the system but for all larger $R$ values they appear at the edge, right next to the rough boundary.

## 3.4 State transitions as a function of off-centered distance $R$

To complete our analysis of the rotational dynamics, we now consider the transition between different rotational states as a function of the off-centered distance $R$ between the center of rotation and the geometrical center of each agent. The full phase diagrams in $D_\theta - R$ plane are shown in Appendix B (Fig. 9). We will focus here on the dynamics represented by the largest angular autocorrelation frequency components.

Despite the differences in vortex dynamics described in previous sections, our angular autocorrelation analysis shows that low and high rotational frequencies are also present in larger systems (see Figs. 12 and 13). Low frequency angular correlations correspond here to the slow drift of all agents around the center of the arena, whereas the high frequency ones still reflect localized rotation. For low $R$ values, localized rotation coexists with collective milling. In small $N = 400$ systems with smooth boundary, they segregate to produce milling in the outer ring and localized rotation in the inner region, separated by a rotating vortex. In larger systems and for a rough boundary, they overlap without a clear structure.

As shown in Figs. 4 and 13 of the paper, low- and high-frequency peaks in the orientation temporal autocorrelation functions can be identified for different values of $R$.

By implementing a peak detection algorithm on the Fourier transformed curves of the orientation autocorrelation, we can display the frequency and amplitude of each peak as a function of $R$, the parameter that controls the off-centered rotation distance, and thus the level of mutual alignment compared to self-alignment.

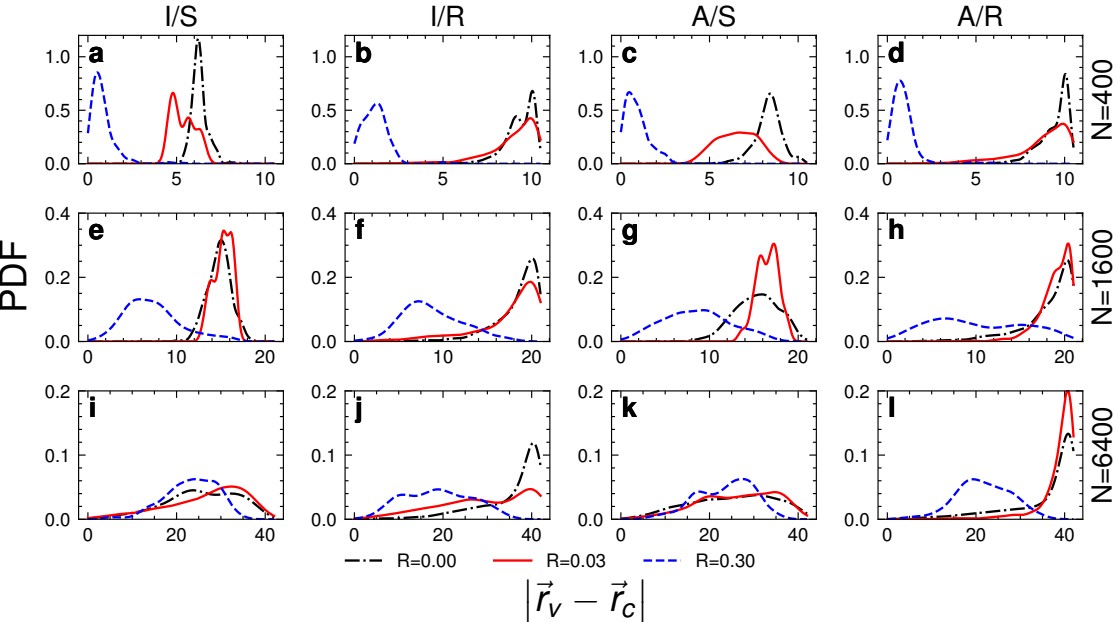

Figure 6: **Probability density functions of vortex radial positions.** We display the PDF of the distance from the center of the arena $\vec{r}_c$ to each identified vortex $\vec{r}_v$ in simulations with smooth (S) or rough (R) arena boundaries, isotropic (I) or anisotropic (A) active disk mobility, different values of $R = 0, 0.03, 0.3$, and different number of disks $N = 400, 1600, 6400$. Each column is labeled by its corresponding combination of I or A mobility and S or R boundaries. The area of arena is increased with $N$ to keep the density constant and each plot spans from the center of the arena to its boundary in the x-axis, thus having a different range for different system sizes. We note that only small systems (with $N = 400$) display a clearly defined central vortex due to milling in the large $R = 0.3$ case, whereas only small and medium size systems (with $N = 400$ and $N = 1600$) with a smooth boundary and $R = 0$ or $R = 0.03$ produce vortices at a fixed distance from the edge of the arena. In all other cases, vortices are continuously appearing and disappearing in different regions of the arena, especially next to its boundary. Other simulation parameters are set to: $l_0 = 1.0$, $v_0 = 0.002$, and $D_\theta = 0.0$.

The Fig. 14 display the frequency and amplitude of the main low- and high-frequency peaks as a function of $R$ for different combinations of isotropic (I) or anisotropic (A) agent-substrate interactions and a smooth (S) or rough (R) confining boundary, as indicated by the column titles. This figure presents the small system case with $N = 400$, whereas Fig. 15 displays large system case with $N = 1600$, keeping the same mean density or packing fraction.

We found that localized rotation is always present for $R < 0.05$, regardless of system size. It is suppressed for higher $R$, where collective milling is favored. We also considered low-frequency milling around the center of the arena. Fig. 14 and 15 reveal that it is always present and that it displays a relatively constant angular momentum for all $R$, with a similar value for all considered setups (isotropic or anisotropic damping, rough or smooth boundary).

## 3.5 State transitions as a function of density

We now explore the transition between dynamical states resulting from changes in the mean density of the system. Up to now, we had always considered the same fixed density $\phi = 0.907$, for which agents are in a strongly crystallized arrangement. Here, we will study systems with

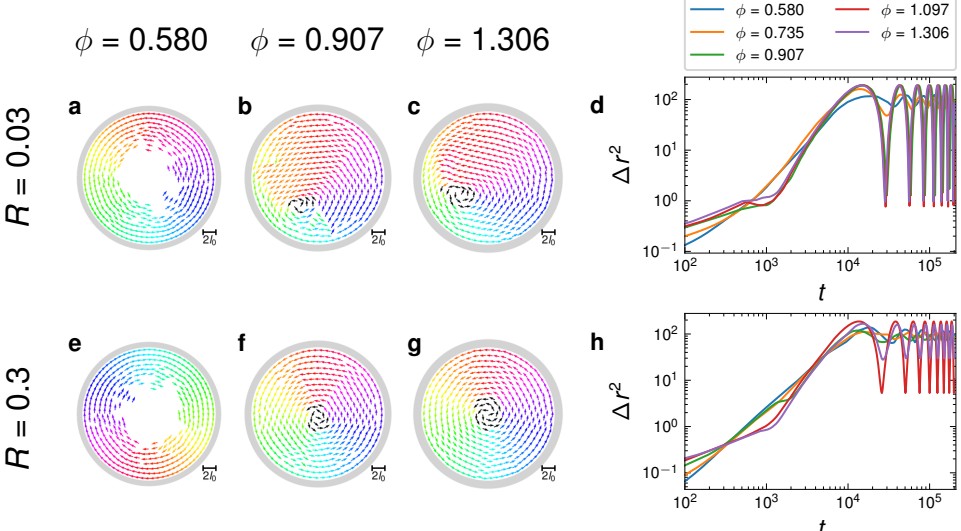

Figure 7: **Emerging collective states and mean squared displacement for different densities and $R$ values, in a smooth boundary confinement.** The packing fraction $\phi$ is controlled by the disk diameter $l_0$ in a system of $N = 400$ disks, setting $l_0 = 0.8$, $\phi = 0.580$ in panels **a**, **e**; $l_0 = 1.0$, $\phi = 0.907$ in **b**, **f**; and $l_0 = 1.2$, $\phi = 1.306$ in **c**, **g**. The corresponding polar ($P$) and milling ($M$) order parameters are given by: $P = 0.012$, $M = 0.997$ in **a**; $P = 0.387$, $M = 0.753$ in **b**; $P = 0.308$, $M = 0.821$ in **c**; $P = 0.015$, $M = 0.998$ in **e**; $P = 0.044$, $M = 0.977$ in **f**; and $P = 0.005$, $M = 1$ in **g**. Panels **d** and **h** display the corresponding mean squared displacement of the disks as a function of time for different packing fractions. Panels **d** and **h** display again the corresponding mean squared displacement of the disks for $R = 0.03$ and $R = 0.3$, respectively, as a function of time and for different packing fractions. Other simulation parameters are set to $\mu_{\parallel} = 0.02$ and $\mu_{\perp} = 0$ (imposing anisotropic damping), $v_0 = 0.002$, and $D_{\theta} = 0$.

lower and higher packing fractions, for agents with smaller or larger off-centered rotation distance $R$, with isotropic or anisotropic disk-substrate damping, and in smooth or rough circular confinements.

To carry out our analyses, we compute the mean squared displacement (MSD) of the particle positions as a function of time for simulations with different densities and setups, to compare their corresponding dynamics based on the resulting curves. The mean squared displacement (MSD) quantifies how far a particle has moved from its initial position over time, providing insights into its diffusive behavior within a system. The MSD for $N$ particles with position vectors $\vec{r}_i(t)$ is defined by [40]

$$\Delta r^2(t) = \frac{1}{N} \sum_{i=1}^{N} [\vec{r}_i(t) - \vec{r}_i(0)]^2 .  \tag{6}$$

Here, $\vec{r}_i(0)$ and $\vec{r}_i(t)$ are the position vectors of the $i$-th particle at initial time $t = 0$ and at a later time $t > 0$, respectively. As in the orientation autocorrelation case, we set the time $t$ to zero after reaching a stationary state, and then compute the $\Delta r^2(t)$ as a function of $t$.

In our confined system, the MSD always saturates at a finite level, but it can still help us distinguish between milling or oscillating, solid or fluid states, as shown in Figs. 7 and 8.

We change the density by varying the disk diameter $l_0$ while keeping constant the number of agents $N = 400$ and the size of the arena $R_c = 10.5$. We study two types of boundary conditions, analyzing simulations in arenas with smooth or rough circular confinements, while

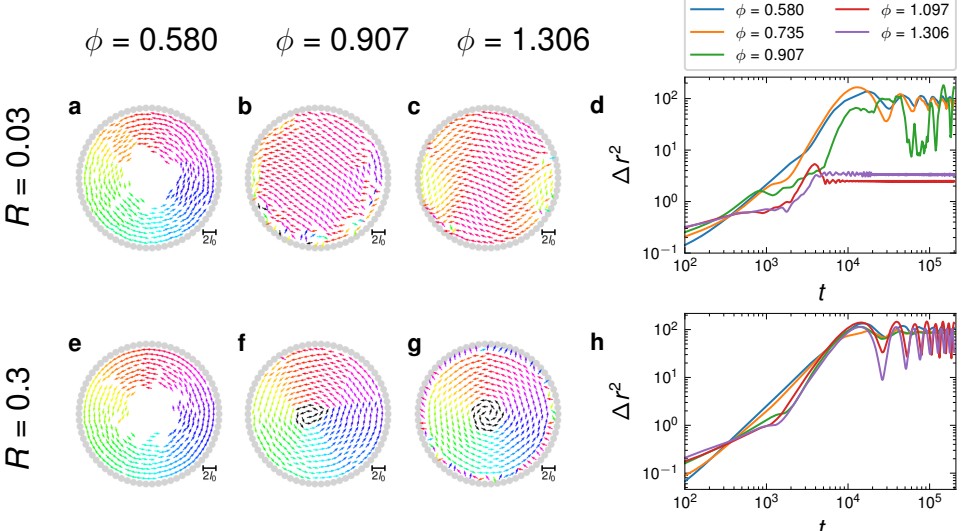

Figure 8: **Emerging collective states and mean squared displacement for different densities and *R* values, in a rough boundary confinement.** As in Fig. 7, the packing fraction $\phi$ is controlled by the disk diameter $l_0$ in a system of $N = 400$ disks, setting $l_0 = 0.8$, $\phi = 0.580$ in **a**, **e**; $l_0 = 1.0$, $\phi = 0.907$ in **b**, **f**; and $l_0 = 1.2$, $\phi = 1.306$ in **c**, **g**. The corresponding polar ($P$) and milling ($M$) order parameters are given by: $P = 0.017, M = 0.990$ in **a**; $P = 0.585, M = 0.396$ in **b**; $P = 0.835, M = 0.065$ in **c**; $P = 0.023, M = 0.996$ in **e**; $P = 0.107, M = 0.933$ in **f**; and $P = 0.098, M = 0.799$ in **g**. Panels **d** and **h** display again the corresponding mean squared displacement of the disks for $R = 0.03$ and $R = 0.3$, respectively, as a function of time and for different packing fractions. Other simulation parameters are set to $\mu_{\parallel} = 0.02$ and $\mu_{\perp} = 0$ (imposing anisotropic damping), $v_0 = 0.002$, and $D_{\theta} = 0$.

focusing on anisotropic damping. Finally, we consider two scenarios for the off-centered rotation distance: a small $R = 0.03$ case and a large $R = 0.3$ case, corresponding to weak and strong explicit alignment, respectively. Note that $R$ is defined here as independent of disk diameter, so changing the value of $l_0$ will also change the relative off-centered rotation distance with respect to its radius. We thus find that, for $R = 0.03$ and $l_0 = 0.8, 1.0, 1.2$, we have $R/(l_0/2) = 0.075, 0.06, 0.05 << 1$, all representing weak alignment. For $R = 0.3$, we have $R_r = 0.75, 0.6, 0.5$, which correspond instead to strong alignment.

Fig. 7 displays snapshots and the MSD for simulations in a smooth boundary arena, with different densities given by low ($\phi = 0.580$) and high ($\phi = 0.907$, $\phi = 1.306$) packing fractions. The corresponding temporal dynamics are displayed as videos in the Supplemental Material. Supplementary Movies 9-11 correspond to the snapshots in Figs. 7a-c and Supplementary Movies 12-14 corresponds to the snapshots in Figs. 7e-g. For the small $R = 0.03$ case, Panel (a) shows that, at low densities, we find clear milling motion around an empty center in a state characterized by order parameters $P = 0.012$ and $M = 0.997$. At higher densities, this milling state is suppressed as the central region is filled with disks and the order parameters become $P = 0.387$ and $M = 0.753$ in Panel (b) and $P = 0.308, M = 0.821$ in Panel (c), which corresponds more to an aligned state displaying localized rotation. On the other hand, for the large $R = 0.3$ case in Panels (e-f) the milling motion remains a stable phase for all densities considered, with $P = 0.015, M = 0.998$ in Panel (e); $P = 0.044, M = 0.977$ in Panel (f); and $P = 0.005, M = 1.000$ in Panel (g). This shows the robustness of the local polarization dynamics driven by strong explicit alignment interactions.

In Panels (d) and (h) of Fig. 7 we display the MSD curves for all the simulations presented in the figure snapshots. Here, the curves must saturate at the size of the arena, which determines the maximum agent displacement. We note that the low density curves tend to saturate at a relatively constant $\Delta r^2$ values, whereas the high density curves show stronger, large amplitude oscillations that periodically decay to close to zero. The former case corresponds to a rotating fluid state, where the MSD reaches a maximum given by the size of the arena, while the latter corresponds to a rotating solid, where all disks return to almost their original position after one milling orbit. In this context, we note that a more perfect solid rotation appears to be achieved in the small $R = 0.03$ case, where the oscillations reach lower $\Delta r^2$ values.

Fig. 8 presents the snapshots and MSD for simulations with a rough boundary. The corresponding temporal dynamics are displayed as videos in the Supplemental Material. Supplementary Movies 15-17 correspond to the snapshots in Figs. 8a-c and Supplementary Movies 18-20 corresponds to those in Figs. 8e-g. For small $R = 0.03$, at low density $\phi = 0.580$, we find again strong milling, with $P = 0.017$, $M = 0.990$, and an empty central region, as displayed in Panel (a). For higher densities, we observe increasing polarization in Panels (b) and (c), with $P = 0.585$, $M = 0.396$ for $\phi = 0.907$, and $P = 0.835$, $M = 0.065$ for $\phi = 1.306$. These are rotating polarized states formed by agents displaying localized rotation. For large $R = 0.3$, as in the smooth boundary case, the milling state is much more robust and remains stable for all densities, with $P = 0.023$, $M = 0.996$ for $\phi = 0.580$ in Panel (e); $P = 0.107$, $M = 0.933$ for $\phi = 0.907$ in Panel (f); and $P = 0.098$, $M = 0.799$ for $\phi = 1.306$ in Panel (g).

In Panels (d) and (h) of Fig. 8 we display the MSD curves for the corresponding figure snapshots. Here again, the curves saturate due to the finite size of the arena, while oscillations that reach low values of $\Delta r^2$ reflect dynamics close to solid-body rotation around the center of the arena. For low $R = 0.03$, the system rotates in a fluid state around the walls, while vacating its center. As the density is increased to $\phi = 0.907$, we note that $\Delta r^2$ reaches lower values, showing that for some periods of time, the system rotates as a solid body. For higher $\phi = 1.097$ and $\phi = 1.306$ densities, $\Delta r^2$ drops significantly, which shows the transition to localized rotation with no milling around the center of the arena. By contrast, for large $R = 0.3$, this rough boundary case behaves very similar to the smooth boundary one, with somewhat smaller oscillations due to the perturbations introduced by the rough boundary. In some cases, the high density dynamics displayed in Panel (g) switches to localized collective rotation, but this state is unstable and reverts to milling after a short period of time.

In sum, at a low $\phi = 0.58$ density, we always find fluid-like milling around an empty circular central region, for a rough or smooth boundary and high or low $R$. High $\phi = 1.306$ density favors instead solid body dynamics, either as solid-body milling or as localized rotation. Rough boundaries tend to introduce defects in the solid structure. Interestingly, they also pin the rotational drift for low $R = 0.03$ and $\phi = 0.907$ or $\phi = 1.306$, resulting in pure localized rotation, and therefore in a low $\Delta r^2$.

# 4 Conclusion

We studied a minimal model of active polar disks with self-alignment in their dynamical equations and mutual alignment resulting from torques introduced by their off-centered rotation. We considered multiple model variations, including isotropic or anisotropic mobility, rough or smooth confinement, and different system sizes and densities. We identified a variety of emerging collective state with different structures and dynamics, combining polarized regions, milling, and vortex formation.

Despite the observed diversity of emerging states, they all display a combination of slow rotational frequencies associated to milling and high rotational frequencies linked to local-

ized circular oscillations. The milling dynamics is mainly driven by mutual alignment forces that increase with $R$, resulting from a decentralized consensus process on the orientation that produces local polarization leading to collective rotation due to the circular confinement. The localized circular oscillations, on the other hand, are driven by self-alignment and its tendency to excite low collective oscillatory modes, here corresponding to the combined horizontal and vertical oscillations that produce localized circular motion. In this framework, the different emergent states reflect the combined effect of self- and mutual alignment as a function of $R$, which can produce solid or fluid regimes in ordered or disordered configurations, depending on the specific setup and parameters. Their analysis thus opens the door to a better understanding of these two self-organizing mechanisms in active matter.

Our simulations demonstrate that the mechanical model of a dense system of active polar disks robustly produces a superposition of collective milling and localized oscillations across various scenarios, including self-aligning dynamics, off-centered rotation distances, isotropic or anisotropic disk-substrate damping, and boundary conditions. The implementation details of the model appear not to affect the emergence of these collective states. Thus, this minimal model captures the generic combination of states that can be realized in dense active elastic matter, integrating the effects of self-alignment and mutual alignment. It could therefore be used to describe a variety of real-world biological and robotic active systems.

# Acknowledgments

We are grateful to Silke Henkes for helpful discussions.

**Funding information**    W.T. acknowledges the National Natural Science Foundation of China, Grant 62176022. Y.Z. and P.R. thank the support of Deutsche Forschungsgemeinschaft (DFG, German Research Foundation) under Germany's Excellence Strategy-EXC 2002/1 'Science of Intelligence'(SCIoI), Project 390523135. The work of C.H. and A.S. was supported by the John Templeton Foundation, Grant 62213.

**Author contributions**    Y.Z. and W.T. performed the numerical simulations and generated the plots. All authors helped contextualize the results and contributed to the writing of the manuscript.

# A    Agent-based simulation timestep

We implement a Euler method to integrate the position dynamics following equation (1). The explicit timestep for the $i$-th agent position is given by

$$x_i(t+dt) = x_i(t) + v_0 \cos(\theta_i)dt + \mu_\parallel \left[f_i^x \cos(\theta_i) + f_i^y \sin(\theta_i)\right]\cos(\theta_i)dt$$
$$- \mu_\perp \left[-f_i^x \sin(\theta_i) + f_i^y \cos(\theta_i)\right]\sin(\theta_i)dt, \tag{A.1}$$
$$y_i(t+dt) = y_i(t) + v_0 \sin(\theta_i)dt + \mu_\parallel \left[f_i^x \cos(\theta_i) + f_i^y \sin(\theta_i)\right]\sin(\theta_i)dt$$
$$+ \mu_\perp \left[-f_i^x \sin(\theta_i) + f_i^y \cos(\theta_i)\right]\cos(\theta_i)dt. \tag{A.2}$$

We use a Euler-Maruyama method to integrate the orientation dynamics in equation (2) and write an explicit timestep for the angular dynamics, given by

$$\theta_i(t+dt) = \theta_i(t) + \beta\left[-f_i^x \sin(\theta_i) + f_i^y \cos(\theta_i)\right]dt + \sqrt{2D_\theta dt}\,\eta_i(t). \tag{A.3}$$

Table 1: Parameters and default values.

| Parameter | Default value | Definition |
|---|---|---|
| $N$ | 400, 1600, 6400 | Total number of particles |
| $l_0$ | 1 | Disk diameter |
| $R_c$ | 10.5 | Radius of the arena |
| $\phi = N l_0^2 / 4 R_c^2$ | 0.907 | Packing fraction |
| $v_0$ | 0.002 | Active speed |
| $D_\theta$ | 0.0 to 0.1 | Noise strength |
| $k$ | 5 | Elastic coefficient |
| $\beta$ | 1.2 | Force alignment strength |
| $R$ | 0.0 to 0.5 | Off centered distance |
| $\mu_\parallel$ | 0.02 | Parallel mobility coefficient |
| $\mu_\perp$ | 0.00 or 0.02 | Perpendicular mobility coefficient |
| $dt$ | 0.01 | Simulation time step |
| $T$ | $2 \times 10^7$ | Total number of time steps |
| $T \times dt$ | $2 \times 10^5$ | Total simulation time |
| $n_{\text{rep}}$ | 5 | Number of simulation repetitions |

We derive the equations for the isotropic and anisotropic limit cases considered in the paper. For isotropic disk-substrate damping, $\mu \equiv \mu_\parallel = \mu_\perp$, and equations (A.1) and (A.2) simplify to

$$x_i(t+dt) = x_i(t) + v_0 \cos(\theta_i) dt + \mu f_i^x dt, \tag{A.4}$$

$$y_i(t+dt) = y_i(t) + v_0 \sin(\theta_i) dt + \mu f_i^y dt. \tag{A.5}$$

For anisotropic damping, we study the extreme anisotropy case where $\mu_\perp = 0$, with no possible displacement perpendicular to the orientation, as in the case of wheeled vehicles. In this case, equations (A.1) and (A.2) simplify to

$$x_i(t+dt) = x_i(t) + v_0 \cos(\theta_i) dt + \mu_\parallel \left[ f_i^x \cos(\theta_i) + f_i^y \sin(\theta_i) \right] \sin(\theta_i) dt, \tag{A.6}$$

$$y_i(t+dt) = y_i(t) + v_0 \sin(\theta_i) dt + \mu_\parallel \left[ f_i^x \cos(\theta_i) + f_i^y \sin(\theta_i) \right] \sin(\theta_i) dt. \tag{A.7}$$

A summary of all the simulation variables and their corresponding values is provided in Table 1. All our results are provided in natural units defined by the equations of motion and the parameter values provided above. For example, since we defined $l_0 = 1$, the unit of length in all our results is given by the disk diameter.

# B  Phase diagrams

Figure 9 presents exploratory phase diagrams in a two-dimensional parameter space, as a function of the off-centered rotation distance $R$ and noise $D_\theta$. We display the polarization $P$ and milling order parameter $M$ as heat maps for the four combinations of isotropic or anisotropic disk-substrate interactions, and of smooth or rough boundaries, that we consider in this work. In addition to these diagrams, Fig. 10 plots the values of $P$ and $M$ as a function of $R$ for low and intermediate noise levels.

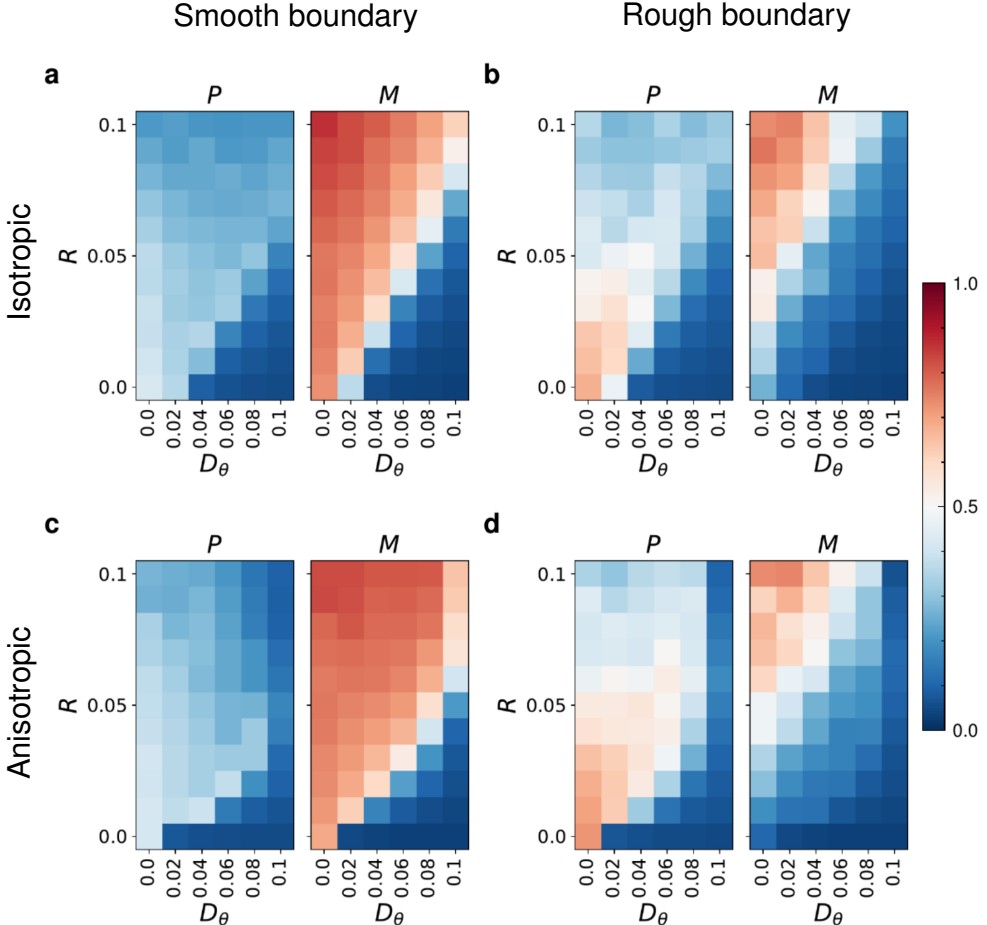

Figure 9: **State Diagrams in the $D_\theta$–$R$ plane.** Polarization $P$ and milling order parameter $M$ as a function of $D_\theta$ and $R$, showing the different collective states that can emerge in systems with isotropic or anisotropic agent-substrate damping and with a rough or smooth boundary. Regions with low $P$ and high $M$ correspond to milling states, high $P$ and low $M$ corresponds to localized rotation states, and high $P$ and $M$ corresponds to disordered states. Here, $l_0 = 1.0$, $v_0 = 0.002$, and each point is averaged over 5 realizations.

These figures show that, for a range of low noise levels, we typically find either regions with low $P$ and high $M$, corresponding to the milling regime, or with high $P$ and low $M$, corresponding to localized rotation. This suggests that these states can emerge in real-world systems with sufficiently low levels of noise, which are the focus of this work. For high noise values, we instead mainly find states with low $P$ and low $M$ that display either quenched or dynamic disorder [55].

## C   Vortex tracking

In order to determine the vortex positions, we find regions of high local vorticity and then compute the mean position of all agents within each of these regions. The vortex location is thus given by $\vec{r}_v = \sum_{i=1}^{N_v} \vec{r}_i / N_v$, where the sum is performed over the $N_v$ position vectors $\vec{r}_i$ of all agents within the high vorticity region. This method is illustrated in Fig. 11.

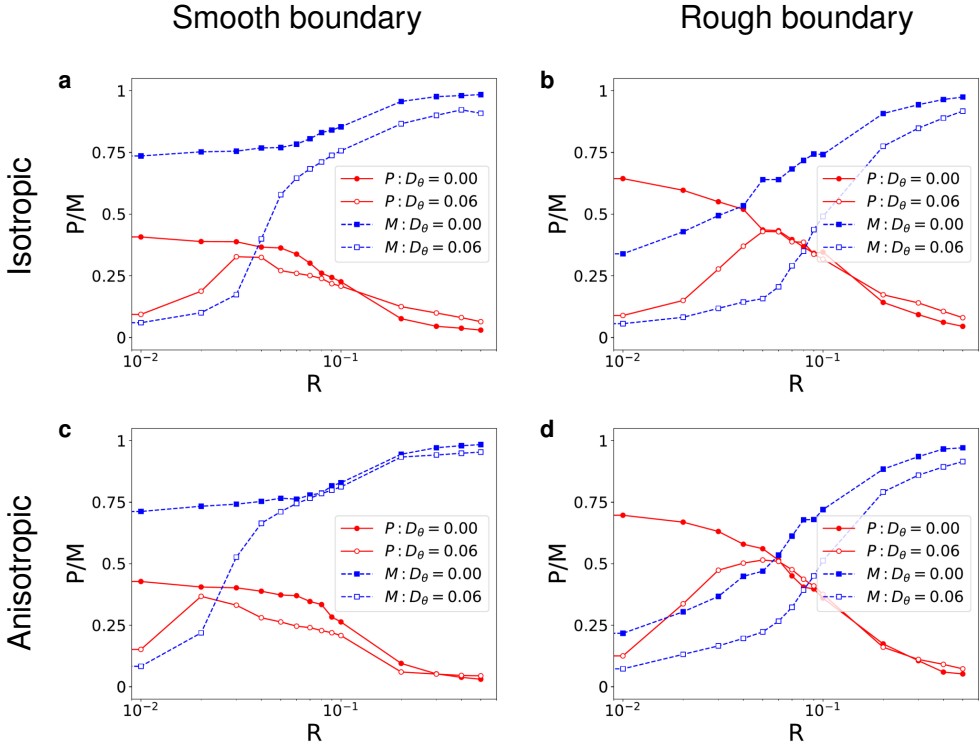

Figure 10: **Polarization and milling as a function of control parameter R.** Polarization $P$ and milling order parameter $M$ as a function of $R$ in systems with isotropic or anisotropic agent-substrate damping and with a rough or smooth boundary. Here, $l_0 = 1.0$, $v_0 = 0.002$, and each point is averaged over 20 realizations.

To analyze the vortex dynamics, we calculate the vortex angular speed by sampling its position in polar coordinates and computing the mean of $\Omega_v$, evaluated every 100 computational time units.

# D    Results in large systems

Fig. 12 presents the orientation autocorrelation functions for the same parameter combinations presented in Fig. 3 of the main paper, but for a larger system with $N = 1600$. Fig. 13 displays the Fourier transforms of these correlation functions, corresponding to the same cases shown in Fig. 4 of the main article, but for a larger $N = 1600$.

Both figures show that the low- and high-frequency oscillations observed in each case are equivalent to those described in the paper for smaller systems.

Fig. 15 display the frequency and amplitude of the main low- and high-frequency peaks as a function of $R$ for different combinations of isotropic (I) or anisotropic (A) agent-substrate interactions and a smooth (S) or rough (R) confining boundary, as indicated by the column titles. Fig. 15 displays large system case with $N = 1600$, keeping the same mean density or packing fraction with Fig. 14.

The figures show that, the high-frequency circular oscillations can always be found at small values of $R$, while the low-frequency milling appears at all $R$ values. This is true for all simulation conditions and for both considered system sizes, $N = 400$ and $N = 1600$.

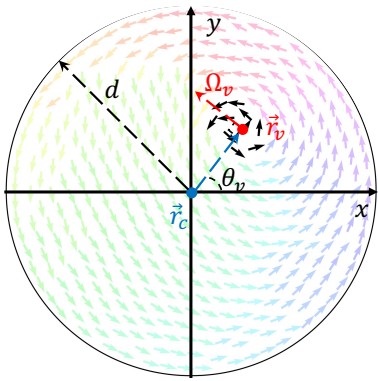

Figure 11: **Tracking of vortex dynamics.** Starting from the orientation of all disks, regions of high vorticity are detected (labeled by black arrows), and the vortex position $\vec{r}_v$ is defined as the mean position of all the agents within this region. This position is then expressed in polar coordinates with respect to the center of the arena $\vec{r}_c$, in terms of the radial position $|\vec{r}_v - \vec{r}_c| \leq d$ and angle $\theta_v$.

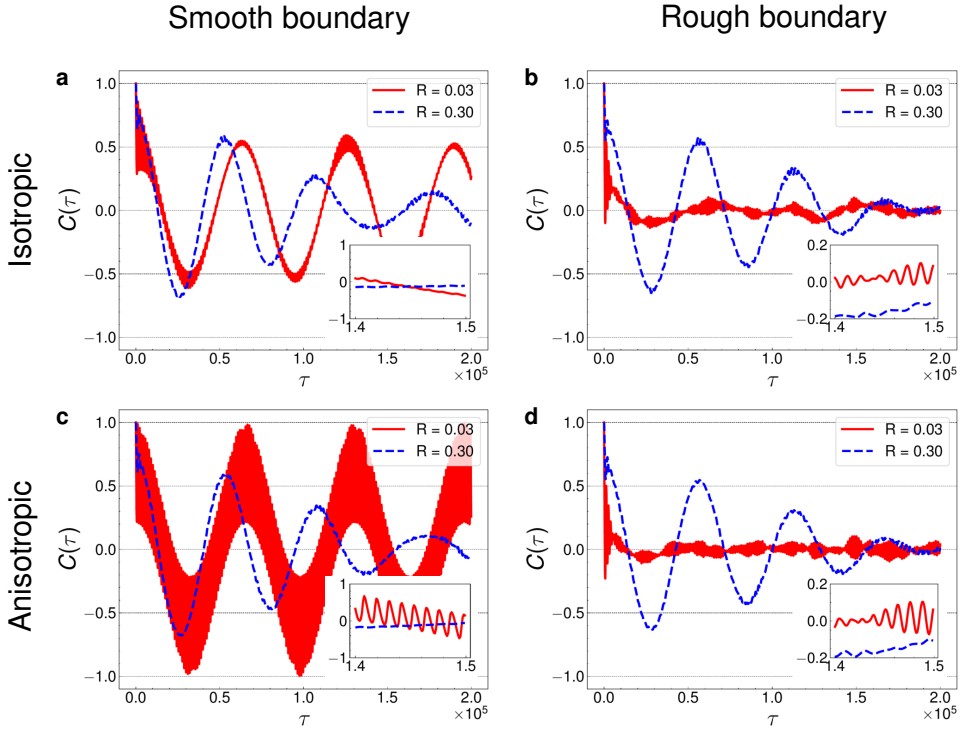

Figure 12: **Orientation temporal autocorrelation functions.** The displayed plots are equivalent to those presented in Fig. 3 of the main paper, but here for a larger system with $N = 1600$ and the same packing fraction. Other simulation parameters are set to: $l_0 = 1.0$, $v_0 = 0.002$, and $D_\theta = 0$.

## E  Vortex dynamics

In the next four figures, we present scatter plots of the locations where vortices appear during a simulation run, for cases with isotropic or anisotropic mobility and smooth or rough circular confinement. In each figure the identified vortex positions are displayed as red dots for different off-centered distances $R = 0.0$, $0.03$, $0.3$, and different system sizes $N = 400$, $1600$,



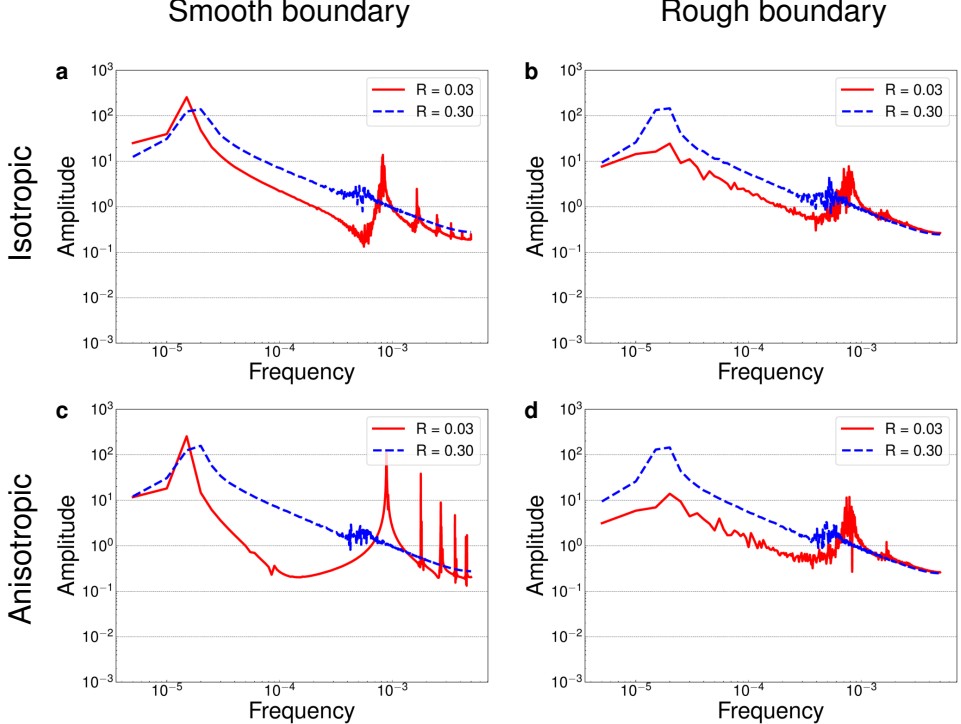

Figure 13: **Fourier transforms of orientation autocorrelation functions.** The displayed plots are equivalent to those presented in Fig. 4 of the main paper, for a larger system with $N = 1600$ and the same packing fraction. Other simulation parameters are set to: $l_0 = 1.0$, $v_0 = 0.002$, and $D_\theta = 0$.

6400, while keeping the packing fraction $\phi$ constant. The number indicated in each panel represents the mean number of vortices identified per snapshot. The cases where this number is approximately equal to 1 typically display either a single milling vortex at the center of the arena or a single orbiting vortex separating the outer milling and inner oscillating regions.

More specifically, Fig. 16 shows cases with isotropic mobility and a smooth confining boundary, Fig. 17 shows cases with isotropic mobility and a rough confinement, Fig. 18 shows cases with anisotropic mobility and smooth confinement, and Fig. 19 shows cases with anisotropic mobility and rough confinement.

We observe, in general, that larger systems tend to have more vortices, but with less structured positions and trajectories. In the smooth boundary case, vortices form mainly in the bulk. In the localized rotation regime, they form a ring of vortices that matches the inner and outer global vorticity in small systems while moving throughout the arena in large systems. In the rough boundary case, vortices mainly nucleate near the edge of the system and then move into the bulk. Finally, we note that for larger system sizes the vortex nucleation and spatial dynamics become increasingly chaotic.

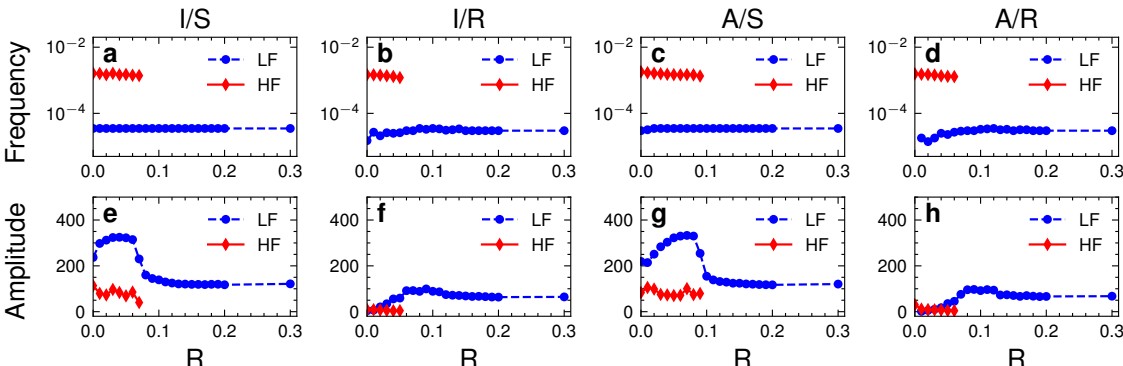

Figure 14: **Frequencies and amplitudes of orientation autocorrelation peaks for** $N = 400$. The low-frequency (LF) and high-frequency (HF) oscillation peaks are identified in the Fourier transform of the orientation autocorrelation function, and their frequency and amplitude is displayed as a function of the off-centered rotation distance $R$. The frequency and amplitude plots are displayed for four different cases of isotropic (I) or anisotropic (A) agent-substrate mobility and smooth (S) or rough (R) boundary. We present the I/S case in panels **a,e**; the I/R case in **b,f**; the A/S case in **c,g**; and the A/R case in **d,h**. Other simulation parameters are set to: $l_0 = 1.0$, $v_0 = 0.002$, and $D_\theta = 0$.

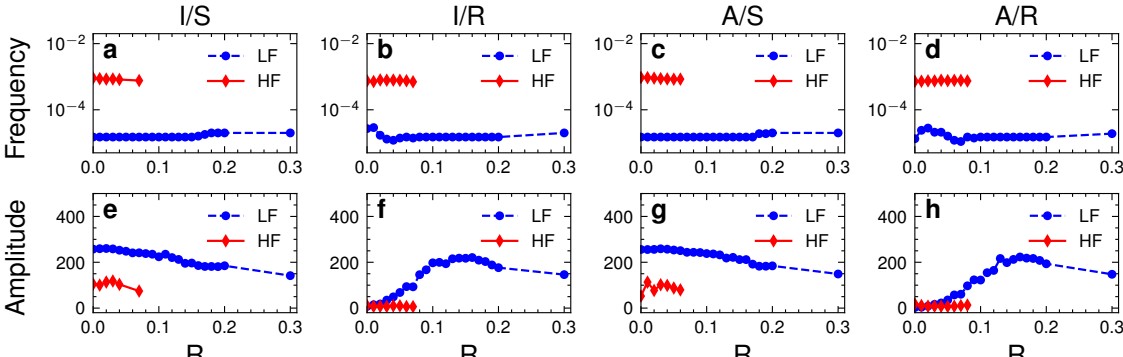

Figure 15: **Frequencies and amplitudes of orientation autocorrelation peaks for** $N = 1600$. The low-frequency (LF) and high-frequency (HF) oscillation peaks are identified in the Fourier transform of the orientation autocorrelation function, and their frequency and amplitude is displayed as a function of the off-centered rotation distance $R$. The frequency and amplitude plots are displayed for four different cases of isotropic (I) or anisotropic (A) agent-substrate mobility and smooth (S) or rough (R) boundary. We present the I/S case in panels **a,e**; the I/R case in **b,f**; the A/S case in **c,g**; and the A/R case in **d,h**. Other simulation parameters are set to: $l_0 = 1.0$, $v_0 = 0.002$, and $D_\theta = 0$.

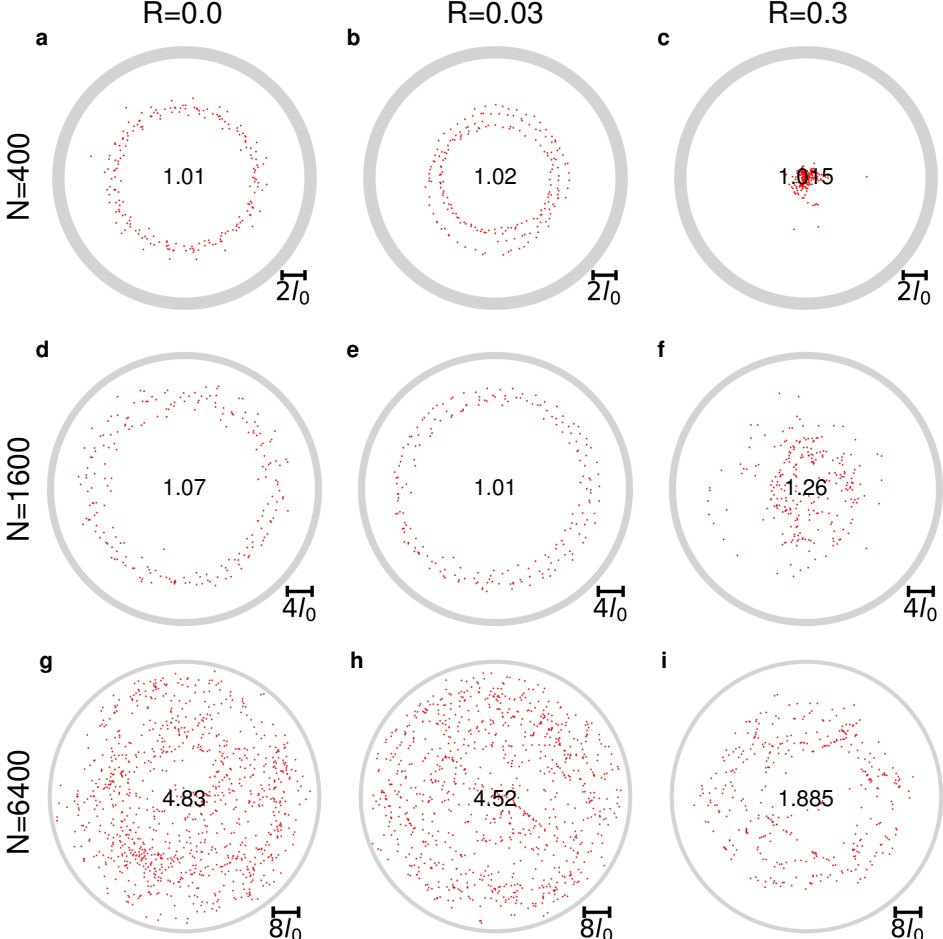

Figure 16: **Vortex positions for isotropic mobility and a smooth boundary.** The red points indicate the vortex positions identified during a run for different off-centered rotation distances and system sizes, keeping the density constant. The number in each panel displays the mean number of vortices per snapshot. Here, agent-substrate interactions are isotropic and the confining boundary is smooth. Other simulation parameters are set to: $l_0 = 1.0$, $v_0 = 0.002$, and $D_\theta = 0.0$.

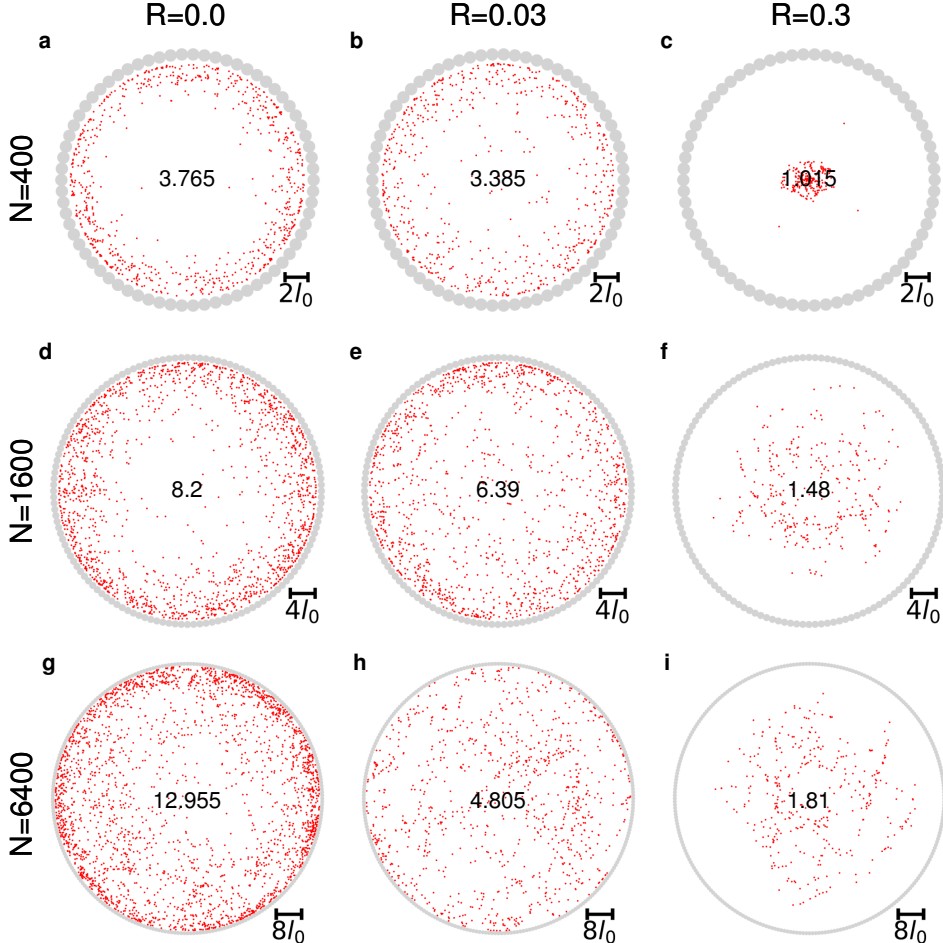

Figure 17: **Vortex positions for isotropic mobility and a rough boundary.** The red points indicate the vortex positions identified during a run for different off-centered rotation distances and system sizes, keeping the density constant. The number in each panel displays the mean number of vortices per snapshot. Here, agent-substrate interactions are isotropic and the confining boundary is rough. Other simulation parameters are set to: $l_0 = 1.0$, $v_0 = 0.002$, and $D_\theta = 0.0$.

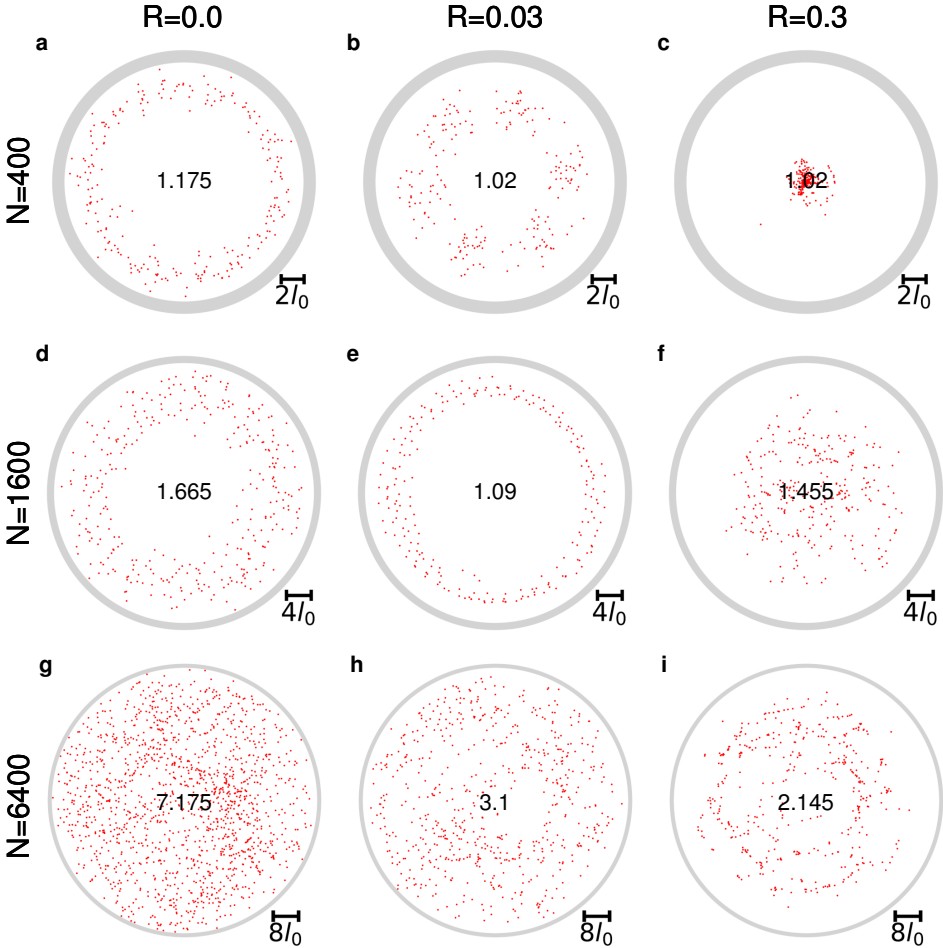

Figure 18: **Vortex positions for anisotropic mobility and a smooth boundary.**
The red points indicate the vortex positions identified during a run for different off-centered rotation distances and system sizes, keeping the density constant. The number in each panel displays the mean number of vortices per snapshot. Here, agent-substrate interactions are anisotropic and the confining boundary is smooth. Other simulation parameters are set to: $l_0 = 1.0$, $v_0 = 0.002$, and $D_\theta = 0.0$.

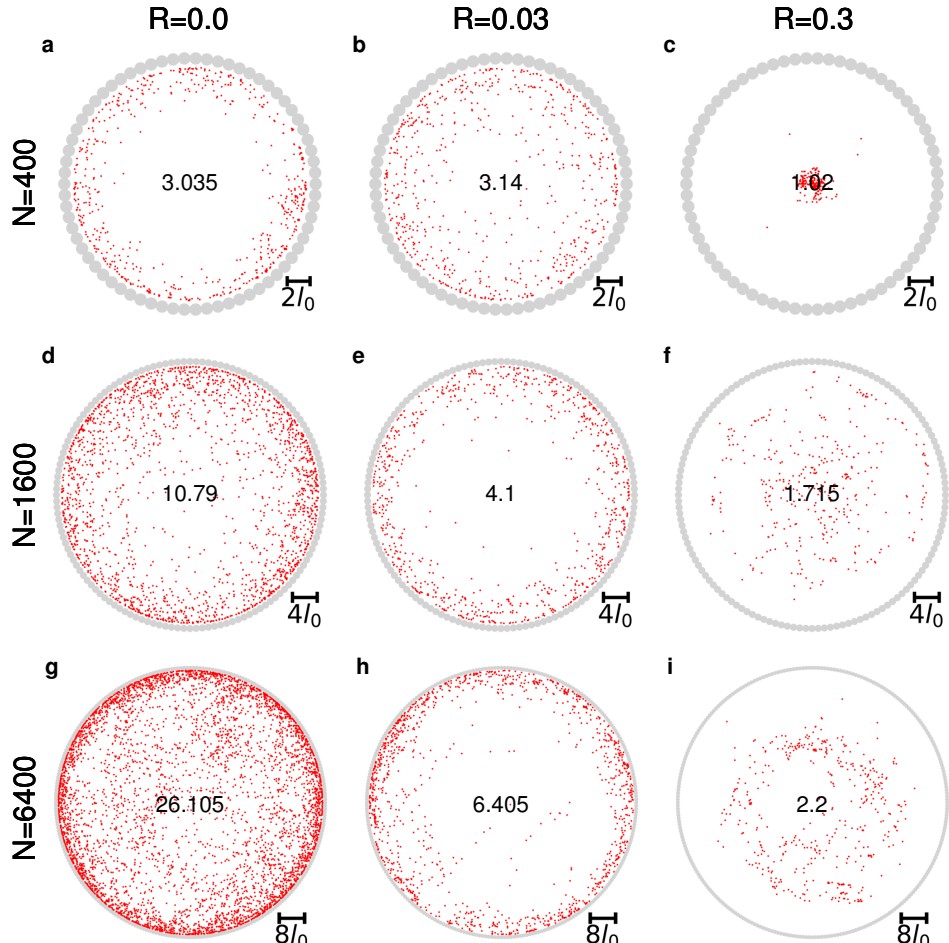

Figure 19: **Vortex positions for anisotropic mobility and a rough boundary.** The red points indicate the vortex positions identified during a run for different off-centered rotation distances and system sizes, keeping the density constant. The number in each panel displays the mean number of vortices per snapshot. Here, agent-substrate interactions are anisotropic and the confining boundary is rough. Other simulation parameters are set to: $l_0 = 1.0$, $v_0 = 0.002$, and $D_\theta = 0.0$.

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
