# Peer review of "Collective dynamics of densely confined active polar disks with self- and mutual alignment"

_SciPost Physics, doi:SciPost Phys. 19, 012 (2025)_

## Round 1 · Referee Report · Anonymous (Referee 1) · 2025-3-5

Report

I read the manuscript ‘Collective dynamics of densely confined active polar disks with self- and mutual alignment” by Weizhen Tang and co-workers. The article presents two-dimensional numerical simulations of self-propelling polar disks in a circular confinement. The main goal is to understand the role of alignment in the emergence of specific collective states. This is done cleverly by controlling the distance R between the geometrical center and the center of rotation of the disks, since mutual alignments become prominent as R increases. The roughness of the circular wall and type of damping (isotropic vs. anisotropic) are also considered. The simulations reveal that milling vortices form when mutual alignments are important and/or in the presence of smooth circular confinements.

The numerical results are interesting and represent a significant advance compared to, e.g., Refs. 40-41. However, before recommending publication I suggest the authors to improve their presentation. As several parameters (roughness of the boundary, R, type of damping) affect the collective states, it would be useful to include a Section in which they are considered one by one. For example, the authors could begin from the case [R=0.03, isotropic damping, smooth boundary] and illustrate what happens when one of the parameters changes.

Other remarks:

1) Eq. (2) includes white noise, but it does not seem to be relevant for any of the following results. If this is the case, wouldn’t it be better to neglect it entirely? It is awkward to me that rotational noise is considered, but the translational one is not. Also, the diffusion coefficient D_\theta is not defined.

2) I would increase the font of the symbols in Fig. 1a.

3) It would be instructive to see a plot where M (degree of milling) and P (polarization) are plotted as a function of R. For this, few additional simulations for intermediate values of R are required.

4) Page 8, “We note, in addition, that the amplitude …. when compared to their isotropic counterparts.” Where does the reader see this in Figure 3.

5) In all Figures the axes do not have units. Can the authors clarify?

Recommendation

Ask for minor revision

  • validity: -
  • significance: -
  • originality: -
  • clarity: -
  • formatting: -
  • grammar: -

Author:  Yating Zheng  on 2025-05-01  [id 5432]

(in reply to Report 1 on 2025-03-05)

We provide our reply in the PDF file attached.

Attachment:

report1.pdf

---

## Round 1 · Referee Report · Anonymous (Referee 2) · 2025-3-21

Strengths

The article presents a confined system of active disks characterized not only by self-alignment but also by mutual alignment with their neighbors. The combined effect leads to a rich state diagram with polarized and milling states.
The presented model could be relevant for real systems of drone swarms active solids or wheeled robots.

Weaknesses

The clarity of the manuscript could strongly improve if the authors could plot state diagrams reporting the different emergent states. All suggested changes are reported below.

Report

The manuscript is clearly written and meets the criteria of an article published in SciePost.
I would be very happy to support its publication in the journal once the authors have addressed all requested changes.

Requested changes

1- The authors present a numerical model that could mimic the behaviour of realistic systems characterized by self and mutual interactions. At the end of the introduction they state
“We expect experimental systems to display both types of self-organised dynamics in realistic setups…”
Could the authors better speculate which realistic setups they have in mind?

2- The system presented by the authors presents several emergent states. I think the manuscript would gain clarity if the authors could add figures reporting state diagrams with the different collective states indicated by the polarization and milling order parameter.
For a fixed packing fraction, the authors could report two state diagrams, one for the smooth and another one for the rough boundaries, when varying the damping coefficient and the value of R.

3- The authors simulate the active agents as soft disks. How important is the fact that the interactions are soft? Could the authors have considered harder interactions between disks?

4- The authors should define all variables used in the equations. For instance, D_theta in equation (2)

5- At page 5 the authors refer to figure 6 right after referring to figure 1. I would suggest the authors to display the figures in a subsequent order. When referring to figures in the Supporting Material, could the authors explicitly state the figures are in the Supporting?

6- The authors compute global order parameters, such as the polarization and the degree of milling defined in equations 3 and 4, respectively.
The authors should explicitly indicate the minimum and maximum value they expect for P and M in each emergent state. To understand the degree of polarization or milling.

7- Would it be possible to better understand how the different collective states emerges? Could they consider local instead of global order parameters? Such as a polarization computed on each particle, summing over first neighbors. Or the degree of milling computed on each particle, summing over first neighbors. These local order parameters might help understanding the nucleation of the different emergent states.

8- Concerning the behavior observed in Figure 3, that reports the orientation autocorrelation functions, could the authors state whether there is a different behavior between particles at the boundary with respect to those at the center of the system (away from the boundary)?
Could the authors compute one autocorrelation function for particles at the boundary and another one for particles away from the boundary?

9- Could the authors better study the mechanism of appearance and disappearance of a vortex?

10- In Section 3.4 the authors study state transitions between rotational states as a function of R. Also in this case, state diagrams would help to clarify the phenomena hereby reported.
The same holds for the study of the state transition as a function of density, reported
in section 3.5: state diagrams would help to clarify the phenomena hereby reported.

11- In Section 3.4 the authors study state transitions between rotational states as a function of R. Also in this case, state diagrams would help to clarify the phenomena hereby reported.

12- Concerning localized rotations (at page 12), is it possible to study whether localized rotations are present, and eventually merge, for higher values of R?

13- Have the authors considered the possibility of changing the shape of the confining walls? How much the emergent states are affected by the confinement’s geometry?

14- In the conclusions, the sentence starting with “The implementations details…” is repeated twice.

Recommendation

Publish (easily meets expectations and criteria for this Journal; among top 50%)

  • validity: top
  • significance: top
  • originality: top
  • clarity: high
  • formatting: excellent
  • grammar: perfect

Author:  Yating Zheng  on 2025-05-01  [id 5433]

(in reply to Report 2 on 2025-03-21)

We provide our detailed reply in the attached PDF file.

Attachment:

report2.pdf

---

## Round 2 · Referee Report · Anonymous (Referee 1) · 2025-5-19

Report

The revised version of the manuscript addresses the concern raised by the Reviewers. Readability has also improved. I now recommend publication of the article as is.

Recommendation

Publish (easily meets expectations and criteria for this Journal; among top 50%)

---

## Round 2 · Referee Report · Anonymous (Referee 2) · 2025-6-6

Report

The authors have properly addressed all issues raised by the referees.
I have read the revised version of the manuscript and it has improved accordingly to the referees' requests.
For these reasons, I am inclined to recommend publication of the manuscript as is.

Requested changes

No further changes requested

Recommendation

Publish (surpasses expectations and criteria for this Journal; among top 10%)

---

## Round 2 · Author Response

Dear Editors,

Thank you for reviewing and handling our submission ‘Collective dynamics of densely confined active polar disks with self- and mutual alignment’(manuscript ID: scipost 202501 00037v1).

We are resubmitting a revised version, following the reviewers' suggestions. We have highlighted our changes to the manuscript in red, and our responses below in blue.

We believe that our manuscript has greatly improved as a result of the reviewers’ feedback. All comments, questions, and criticisms have been addressed below. We thus respectfully request that our manuscript be accepted for publication in Scipost.

On behalf of all the authors, With regards,
Pawel Romanczuk and Cristián Huepe

---

## Editorial Decision

published